

# No relevant differences in conditioned pain modulation effects between parallel and sequential test design. A cross-sectional observational study

Roland R. Reezigt[1,2], Sjoerd C. Kielstra[1], Michel W. Coppieters[1,3] and Gwendolyne G.M. Scholten-Peeters[1]

[1] Department of Human Movement Sciences, Faculty of Behavioural and Movement Sciences, Vrije Universiteit Amsterdam, Amsterdam, Netherlands
[2] Academy of Health, Department of Physiotherapy, Hanze University of Applied Sciences, Groningen, Netherlands
[3] Menzies Health Institute Queensland, Griffith University, Brisbane and Gold Coast, Australia

## ABSTRACT

**Background**. Conditioned pain modulation (CPM) is measured by comparing pain induced by a test stimulus with pain induced by the same test stimulus, either during (parallel design) or after (sequential design) the conditioning stimulus. Whether design, conditioning stimulus intensity and test stimulus selection affect CPM remains unclear.
**Methods**. CPM effects were evaluated in healthy participants ($N = 89$) at the neck, forearm and lower leg using the cold pressor test as the conditioning stimulus. In three separate experiments, we compared the impact of (1) design (sequential versus parallel), (2) conditioning stimulus intensity (VAS 40/100 *versus* VAS 60/100), and (3) test stimulus selection (single versus dual, *i.e.,* mechanical and thermal). Statistical analyses of the main effect of design (adjusted for order) and experiment were conducted using linear mixed models with random intercepts.
**Results**. No significant differences were identified in absolute CPM data. In relative CPM data, a sequential design resulted in a slightly lower CPM effect compared to a parallel design, and only with a mechanical test stimulus at the neck (−6.1%; 95% CI [−10.1 to −2.1]) and lower leg (−5.9%; 95% CI [−11.7 to −0.1]) but not forearm (−4.5%; 95% CI [−9.0 to 0.1]). Conditioning stimulus intensity and test stimulus selection did not influence the CPM effect nor the difference in CPM effects derived from parallel versus sequential designs.
**Conclusions**. Differences in CPM effects between protocols were minimal or absent. A parallel design may lead to a minimally higher relative CPM effect when using a mechanical test stimulus. The conditioning stimulus intensities assessed in this study and performing two test stimuli did not substantially influence the differences between designs nor the magnitude of the CPM effect.

Corresponding authors
Roland R. Reezigt, r.r.reezigt@vu.nl
Gwendolyne G.M. Scholten-Peeters, g.g.m.scholten-peeters@vu.nl

## INTRODUCTION

Conditioned pain modulation (CPM) is of increasing interest in studies assessing central pain mechanisms (*Damien et al., 2018*). CPM is a paradigm that tests the central pain inhibiting mechanism in response to a second nociceptive stimulus (*Pud, Granovsky & Yarnitsky, 2009*; *Yarnitsky et al., 2010*). Patients with diverse pathologies, such as fibromyalgia (*O'Brien et al., 2018*), chronic low back pain (*Rabey et al., 2015*), migraine (*Williams et al., 2019*), carpal tunnel syndrome (*Soon et al., 2017*), Achilles tendinopathy (*Tompra, Van Dieën & Coppieters, 2015*), and many chronic pain conditions (*Lewis, Rice & McNair, 2012*) show a diminished CPM effect compared to healthy participants. This diminished effect, associated with less endogenous analgesia, may underlie persistent pain (*Edwards, 2005*; *Lewis, Rice & McNair, 2012*; *Staud, 2012*; *Van Wijk & Veldhuijzen, 2010*).

For CPM quantification, pain induced by a test stimulus is compared to pain induced by the same test stimulus but applied during or immediately after a painful conditioning stimulus (*Nir et al., 2011*). Different modalities of test stimuli and conditioning stimuli are used to measure CPM (*Kennedy et al., 2016*). Commonly used test stimuli are mechanical (*e.g.*, pressure pain thresholds and cuff pressure algometry (*Klyne et al., 2018*; *Skovbjerg et al., 2017*)), and thermal stimuli (*e.g.*, heat (*Gehling et al., 2016*; *Horn-Hofmann et al., 2018*)). The cold pressor test (*Grouper, Eisenberg & Pud, 2019*; *Schliessbach et al., 2019*) and ischaemic pressure (*Graven-Nielsen et al., 2017*; *Smith & Pedler, 2018*) are frequently used conditioning stimuli. Further, after applying a first test stimulus (the baseline), the second test stimulus can be applied simultaneously with the conditioning stimulus (parallel design) (*Hoegh, Petersen & Graven-Nielsen, 2018*; *Lie et al., 2017*) or following the conditioning stimulus (sequential design) (*Flood et al., 2017*; *Grosen et al., 2014*). Even though many different CPM protocols are currently in use, it is unclear whether these different protocols have a substantial influence on the magnitude of the CPM effect and may partly explain discrepancies between CPM findings reported in the literature. These discrepancies in the CPM effect are observed in healthy participants (*e.g.*, the effect may be anti-nociceptive, pro-nociceptive, or there may be no effect (*Klyne et al., 2015*; *Larsen, Madeleine & Arendt-Nielsen, 2019*; *Mertens et al., 2020*; *Skovbjerg et al., 2017*)). Furthermore, the size of the CPM effect may be inconsistent (*e.g.*, elite athletes may show a smaller, similar or larger CPM effect compared to healthy participants (*Mertens et al., 2020*; *McDougall et al., 2020*), and similar inconsistencies are observed within patient populations (*Lewis, Rice & McNair, 2012*; *Owens et al., 2016*; *Xie et al., 2020*)).

Additionally, practical recommendations to measure the CPM effect have been formulated (*Yarnitsky et al., 2015*). These recommendations suggest using both mechanical and thermal test stimuli within a protocol for comparability, and a sequential design to limit distraction bias while delivering the conditioning stimulus. Interestingly, the CPM effect duration after termination of the conditioning stimulus seems to be relatively short, varying from 0 to 10 min (*Imai et al., 2016*; *Lewis et al., 2012*; *Vaegter, Handberg & Graven-Nielsen, 2016*). Local, regional and remote locations are often selected in clinical studies to differentiate between primary, secondary, or widespread hyperalgesia or gain of nerve

fibre function (*Courtney et al., 2016*; *Nahman-Averbuch et al., 2020*; *Ng et al., 2014*; *Owens et al., 2016*; *Vaegter, Handberg & Graven-Nielsen, 2014*). Therefore, the application of test stimuli on multiple locations takes time, which may justify using a parallel design where the duration of the conditioning stimulus can be extended until the last test stimulus (*Pud, Granovsky & Yarnitsky, 2009*). Little is known however about potential differences in the magnitude of the CPM effect between a sequential or parallel design. Indirect studies reported conflicting results, showing no differences between both designs (*Ibancos-Losada et al., 2020*), larger effects for either a parallel (*Nahman-Averbuch et al., 2013*; *Ram et al., 2008*) or sequential (*Enax-Krumova et al., 2020*) design. Moreover, statistical comparisons between the two designs were often not performed (*Enax-Krumova et al., 2020*; *Nahman-Averbuch et al., 2013*; *Ram et al., 2008*).

Additionally, even though a mild to moderate pain intensity of the conditioning stimulus is sufficient to induce a CPM effect (*Nir et al., 2011*), it is unclear whether the intensity affects the magnitude of the CPM effect differently in the designs when the duration of the conditioning stimulus is different (*Razavi et al., 2014*; *Smith & Pedler, 2018*). Moreover, it is unknown whether the combination of both mechanical and thermal test stimuli, as the recommendations prescribe (*Yarnitsky et al., 2015*), may interfere with each other. One test may lead to skin hyperalgesia which can influence the measurements necessary for the CPM effect (*Møiniche, Dahl & Kehlet, 1993*; *Rgens et al., 2014*).

The choice of which protocol to use in clinical research is often based on the recommendations (*Yarnitsky et al., 2015*), but substantiated deviations could be made. Therefore, this study aimed to (1) assess possible differences in the magnitude of the CPM effect between a sequential and parallel design, using designs for clinical usage according to the recommendations with both a mechanical and thermal stimulus, (2) to evaluate the influence of conditioning stimulus intensity and the use of dual test stimuli (compared to single test stimuli) on (a) potential differences between designs and (b) on the CPM effect itself.

## METHODS

### Study design and planning

A cross-sectional, double-blind observational study with a mixed-design was performed. The study was approved by the scientific and ethical review board of the Vrije Universiteit Amsterdam, The Netherlands (VCWE-2017-022R1). The study was conducted according to the Declaration of Helsinki (2013).

To address the study aims, three separate experiments with different participant groups were conducted. In each experiment, both a sequential and parallel design was performed (as repeated measurement), and the cold pressor test was used as the conditioning stimulus. In Experiment 1, a conditioning stimulus intensity of VAS 40/100 was used, with both mechanical and thermal test stimuli; Experiment 2 was identical to Experiment 1, but the conditioning stimulus intensity was VAS 60/100; Experiment 3 was identical to Experiment 2, but with only a mechanical test stimulus. Table 1 and Fig. 1B provides an overview of the experiments.

**Table 1** Overview of the three experiments.

|  | Experiment 1 | Experiment 2 | Experiment 3 |
| --- | --- | --- | --- |
| Test Stimulus | - Mechanical: Pressure Pain Threshold<br>- Thermal: Heat | - Mechanical: Pressure Pain Threshold<br>- Thermal: Heat | - Mechanical: Pressure Pain Threshold |
| Conditioning Stimulus Intensity (Cold Pressor Test) | VAS 40/100 | VAS 60/100 | VAS 60/100 |

**Notes.**
Overview of experiments with conditioning stimulus intensity and type of test stimulus used in the three experiments. VAS, Visual Analogue Scale.

For the first aim, the difference in CPM effect between parallel and sequential designs was compared across all three experiments. For the second aim, the impact of the conditioning stimulus intensity on CPM was assessed by comparing Experiment 1 and Experiment 2. The impact of using single or dual test stimuli on the CPM effect was evaluated by comparing Experiment 2 and Experiment 3.

The procedures were comparable between the three experiments, consisting of familiarisation, baseline measurements and a sequential and parallel CPM design (see Fig. 1 for an overview of the procedures). Prior to familiarisation, demographic data were collected.

Due to the length of the total procedure, possibly resulting in attention bias, the measurements in Experiment 1 were performed on two consecutive days (the sequential design on one day, the parallel design on another day, in random order). Participants were familiarised with the procedures on the first testing day. All measurements were performed at approximately the same time of the day to account for the possible influence of circadian rhythm (*Imai et al., 2016*). The procedure of Experiment 2 was identical to Experiment 1. Experiment 3 was performed in a single day due to a shorter protocol, with a 20 minutes rest period between the designs.

## Participants

Participants were recruited from the Academy of Health from the Hanze University of Applied Sciences in Groningen, The Netherlands. Participants had to be healthy and pain-free (*Coghill & Yarnitsky, 2015*; *Gierthmühlen et al., 2015*), to avoid influences of health conditions or disorders on pain modulation. Additionally, they had to be naive to pain modulation mechanisms to avoid potential expectation bias (*Nir et al., 2012*). Additionally, due to the potential influence of hand dominance on pain sensitivity (*Pud, Golan & Pesta, 2009*), all participants had to be right-handed. Recruitment was matched in the experimental groups for age and sex to enable comparison (*Granot et al., 2008*; *Hackett, Naugle & Naugle, 2020*; *Leone & Truini, 2019*). Exclusion criteria were: acute pain, a history of chronic pain or chronic pain syndromes, such as fibromyalgia, migraine or irritable bowel syndrome, use of analgesic or psychiatric medication, Reynaud syndrome, intolerance for cold, cardiovascular, respiratory, systematic or neurologic disease, pregnancy or inability to

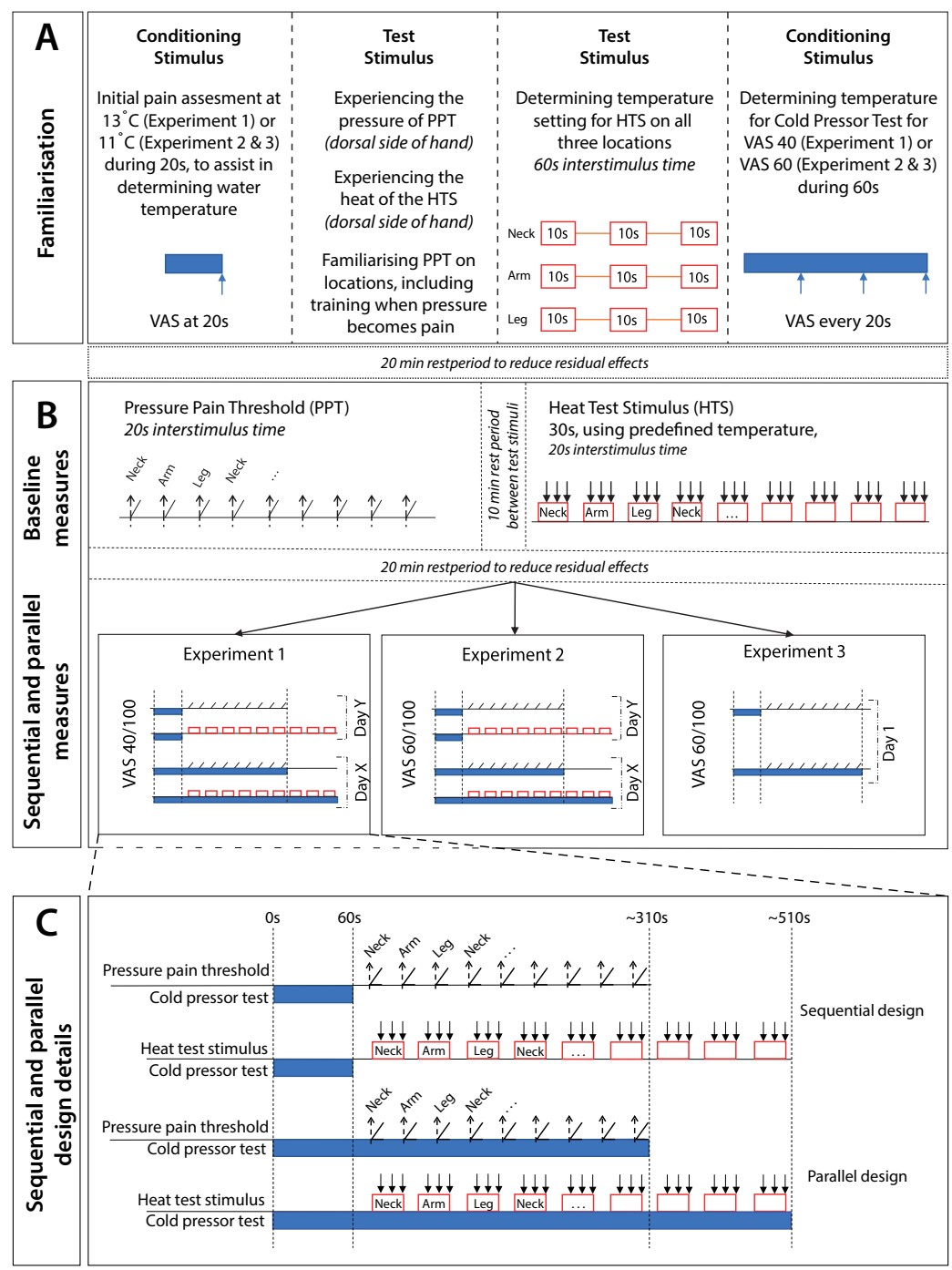

**Figure 1 Overview of the study procedures, including the sequential and parallel design, for both mechanical and thermal test stimuli.** (A) Familiarisation with all stimuli (same for all experiments). (B) Baseline measures and conditioned measures with an overview of all three experiments (N.B. participants were enrolled in only one experiment, where both designs were performed). (C) Detail of the designs. In the sequential design, the conditioning stimulus was terminated after 60 s. The time between testing locations was 20 s. Arrows indicate measurement of VAS every 10 s per heat exposure of 30 s. Test locations; Neck: paraspinal muscles level C5/C6; Arm: extensor carpi radialis muscle; Leg: tibialis anterior muscle.

lie down in a prone position for the duration of the experiment. Furthermore, participants who scored more than the cut-off value of 10 on the Generalised Anxiety and Depression-7 item scale (GAD-7) were excluded (*Arntz, Dreessen & De Jong, 1994*; *Spitzer et al., 2006*). All participants provided written informed consent prior to participating.

Participants were asked to refrain from pain medication, caffeine-containing products (*Baratloo et al., 2016*; *Sawynok, 2011*), alcohol (*Thompson et al., 2017*) and nicotine (*Bagot et al., 2017*; *Girdler et al., 2005*) at least 24 h before and during the measurements to reduce their influence on pain modulation. Participants were kept naïve to the study objective and were asked not to talk about the study with other participants to avoid potential expectation bias (*Nir et al., 2012*). Furthermore, participants could only participate in one of the three experiments.

## Conditioned pain modulation

Mechanical and thermal test stimuli were applied over the right paraspinal muscles at the C5-C6 level, extensor carpi radialis muscle and tibialis anterior muscle as these are frequently used test locations for CPM (*Klyne et al., 2015*; *Neziri et al., 2013*; *Scott, Jull & Sterling, 2005*). The cold pressor test was used as the conditioning stimulus. The left hand (with spread fingers) was submerged in cold water, at an individualised temperature. Participants were positioned in a prone position on a treatment plinth in a temperature-controlled and quiet room. The right arm was placed above the head with the elbow flexed. The right knee was flexed and fixated at the ankle by the rater while testing the lower leg. The prone position was chosen for a better stability compared to a sitting position during the neck and arm measurements.

### *Test stimuli*

As mechanical test stimulus, pressure pain thresholds (PPT) were measured with a digital algometer (Type II, Somedic AB, Stockholm, Sweden) using a 1 cm$^2$ probe with an application rate of 50 kPa/s. The participants pressed the hand-held switch when the feeling of pressure changed into painful pressure (*Rolke et al., 2006*). The PPTs were assessed three times at each test location with an inter-stimulus period of 20 s (*Visscher et al., 2017*).

As thermal test stimulus, a tonic heat test stimulus was delivered, using a 25 mm × 50 mm (12.5 cm$^2$) computer-controlled thermode (MSA Thermal Stimulator, Somedic AB, Stockholm, Sweden). The automatic maximal cut-off temperature was set at 50 °C to prevent possible skin damage (*Rolke et al., 2006*). The method of levels was used at each location to determine the required temperature of the tonic heat test stimulus corresponding with a baseline pain of VAS 40/100 (*Lie et al., 2017*). First, a temperature of 43 °C for 10 s was used and the pain intensity was rated on a VAS. Then, the temperature was adjusted if needed by either 0.5 °C, 1.0 °C or 2.0 °C for the next exposure to achieve the required pain intensity of VAS 40/100. The inter-stimulus period between these exposures was set at 60 s (*Klyne et al., 2015*; *Razavi et al., 2014*). If 50 °C was reached before the pain threshold was reached, 50 °C was used.

During the baseline and conditioned measurements, all three locations were tested three times for 30 s using the temperature obtained, to increase robustness and similarity to

the mechanical stimulus. During each measurement, participants were asked to rate the pain on a VAS scale after 10, 20 and 30 s (*Granot et al., 2008*). The time between the test locations was at least 20 s (*Granot et al., 2008*).

Measurements at the various locations were performed in a fixed cyclic order (*Bisset, Evans & Tuttle, 2015*), starting with the neck and ending with the tibialis anterior muscle. The time between measurement procedures with the mechanical and thermal stimuli during the baseline as well as during the conditioned measurements was 10 min to clear residual effects (*Yarnitsky et al., 2015*). Mechanical and thermal test stimuli were applied in random order. The time between the baseline and conditioned measurements was pragmatically set at 20 min (*Granot et al., 2008*). In Experiment 3, only one baseline measurement was taken. Reasons for this were: the short timeframe in Experiment 3 compared to Experiment 1 and 2 (~1 h *versus* ~24 h), the high reproducibility of the test stimulus (*Arendt-Nielsen et al., 2015*) and prevention of burden for the participants.

### Conditioning stimulus

The cold pressor test was used as conditioning stimulus. The intra-session reliability of measuring CPM with a cold pressure test is good to excellent (ICC = 0.61–0.80) (*Kennedy et al., 2016*). A calibrated, refrigerated circulating water bath with a capacity of 28 litres was used (Polyscience, Illinois, USA). Since the perception of cold is individually different (*Green & Akirav, 2010*), the water temperatures for the cold pressor test were tailored for the individual (*Owens et al., 2016*; *Skovbjerg et al., 2017*), such that VAS 40/100 was reached in Experiment 1 and VAS 60/100 in Experiment 2 and Experiment 3.

To determine the water temperature for each participant, the starting temperature was set at either 13 °C or 17 °C (Experiment 1 with VAS 40/100) and 11 °C or 15 °C (Experiment 2 and Experiment 3 with VAS 60/100), based on the first experience with the cold water during familiarisation (see Fig. 1A). The temperature was decreased at a rate of ~0.2 °C/s until the target VAS was reached. The participants were asked to rate their pain intensity on a VAS every 20 s until the target pain was reached.

### Sequential and parallel design

In the sequential design, the hand was taken out of the cold water after 60 s and dried with a towel by the research assistant (*Perrotta et al., 2010*; *Sandrini et al., 2006*). In the parallel design, the conditioning stimulus started 60 s before applying the test stimulus, and the hand was kept in the water until all measurements of the test stimulus were performed (Fig. 1C).

### Familiarisation

Prior to the measurements on the first testing day, the participants were familiarised with the procedures (Fig. 1A). First, they submerged their left hand in the cold water bath for approximately one minute or until they perceived too much discomfort. Next, both the mechanical and thermal test stimuli were applied on the dorsal side of the right hand to reduce anxiety for the test stimuli (*Rolke et al., 2006*). Subsequently, the participants were familiarised with the pressure pain threshold test procedure, including how to use the hand-held switch. Pressure pain thresholds were applied once on each test location.
Finally, the participants were familiarised with the heat test stimulus procedure and the temperature corresponding to a VAS 40/100 was determined. Standardised instructions were given to all participants. The break time between the familiarisation and the baseline measurements was set pragmatically at 20 min, equal to the break time between baseline and conditioned measurements (*Granot et al., 2008*).

## Randomisation and blinding

Computer-generated block randomisation was used to determine the order of the sequential and parallel design, and the order of the thermal and mechanical test stimulus. An independent person performed the randomisation. The mechanical and thermal test stimuli were applied by two testers (RR and SK) independently of each other and a research assistant applied the conditioning stimulus.

Both testers were blinded to the measurement results and the sequential or parallel design, by using a screen between the treatment plinth and the cold water bath. The research assistant pretended to dry the participant's hand after 60s in the parallel design, behind the screen, to guarantee blinding of the testers.

## Sample size

Sample size calculations were based on a power of $\beta = .80$, a significance level of $\alpha = .05$ and a Generalised Linear Models (GLM) Repeated Measurements approach (*Chi, Glueck & Muller, 2019*; *Muller & Stewart, 2012*), using G*Power 3.1.9.2 (*Faul et al., 2007*). For the main effect between the designs, a medium effect size was chosen ($\eta_p^2 = .09$) as the minimal effect size to be found in the experiments. Consequently, in a $2 \times 3$ mixed-method design of two within-subjects (sequential and parallel) and three between-subjects groups (Experiment 1, 2 and 3), a minimal sample size of 87 participants was needed with at least 29 participants per experiment. Considering an anticipated maximal drop-out rate of ~10% (*Gehling et al., 2016*; *Kjær Petersen, Bjarke Vaegter & Arendt-Nielsen, 2017*), 96 participants were recruited, with 32 participants per experiment.

## Data analysis
### Demographics

We used descriptive analyses to report demographic data. All data, including residuals for analyses of variance (ANOVA), were tested for the assumption of normality using histograms, boxplots, Q-Q plots and the Shapiro–Wilk test. For continuous data with a normal distribution, the mean and standard deviation (SD) were reported. Otherwise, the median and interquartile range (IQR) or percentages were provided. Differences in demographic and baseline data between the three experiments were analysed using Chi-Square, ANOVA or the non-parametric Kruskal-Wallis H-test. Statistical analyses were performed in SPSS version 25 (IBM, Armonk, NY, USA). The significance level for all tests was set at $\alpha = .05$.

### Calculations

The test stimuli results were calculated as the mean of three measurements (*Klyne et al., 2015*). For the mechanical test stimulus, outliers (defined as a 20% difference from the

mean of the other two measurements) were removed, and a fourth measurement was taken (*Tompra, Van Dieën & Coppieters, 2015*). For the heat test stimulus, the mean was calculated of the VAS after 10, 20 and 30 s.

To determine the magnitude of the CPM effect per individual per location, the relative and absolute difference between the mean baseline test and the mean conditioned test were both calculated as the recommendations prescribe (*Yarnitsky et al., 2015*).

Relative CPM effects were calculated with the baseline measurements used as denominator, subtracted by 100%, so the relative CPM effect shows the deviation from the individuals' baseline (percent change). Using this calculation, a result of 0% reflects no difference between the baseline and conditioned pressure pain thresholds, and a positive value reflects the presence of an inhibitory or anti-nociceptive CPM effect. Since the heat test measurement is a patient-reported VAS rating that decreases in case of a CPM effect, we used the same calculation but multiplied it by −1, so that the CPM effects can be interpreted the same way for both types of stimuli.

Absolute differences were calculated by taking the difference of the mean conditioned test and the baseline pain threshold, in concordance with the relative CPM effects, so that positive numbers indicate an inhibiting or anti-nociceptive CPM effect for both relative and absolute data, and negative numbers a facilitatory or pro-nociceptive effect.

### Statistical analyses

Linear mixed model analyses with a Maximum Likelihood (ML) method were used to analyse differences in the magnitude of CPM effect between the sequential and parallel design, with the participants as a random effect. Random slopes were added to the random intercepts in case of significant differences between the models based on -2 log-likelihood tests. Design (sequential or parallel) and experiment (1, 2 and 3) were considered as fixed variables. For the heat test stimulus, only Experiment 1 and Experiment 2 were used. Assumptions of linearity and residual normality were checked before modelling.

First, main effects of design and experiments were calculated per location. Since the designs were randomised, the effect of design was adjusted for the effect of order of measurements. This adjustment separates the effect of which design was used (sequential *versus* parallel) *versus* which design was used first (first *versus* second, irrespective of the design), providing a clearer representation of the main effect of design itself. Additionally, local effect sizes (Cohen's $f^2$, adjusted for order) were calculated for the main effect of design (*Selya et al., 2012*).

Analyses for the effect of experiment were conducted by using dummy variables, with Experiment 2 as control, where the contrast with Experiment 1 reflects the difference in intensity, and the contrast with Experiment 3 reflects the difference in using single or dual test stimulus.

Next, interaction effects between the design and the experiments were calculated, whereas a significant interaction effect reflects significant differences between the sequential and parallel designs due to the experimental conditions. The resulting coefficients, 95% confidence intervals and significance for the main and interaction effects are presented. The significance level for all tests was set at $\alpha = .05$.

Further, estimated marginal means of the absolute and relative magnitude of the CPM effect, including the differences between the sequential and parallel design, are presented in kPa, VAS (0–100) or percentage.

Additionally, to assess the robustness of the results, whether differences between parallel and sequential testing are related to the distribution of the CPM effect itself, a quantile regression analysis was performed (*Cook & Manning, 2013*). Quantile regression analysis enables to analyse beyond the mean of the sample, exploring the whole distribution, including the tails (pro-nociceptive or anti-nociceptive), of the target variable (CPM effect). Regression coefficients for the main effect of design are calculated per location and experiment at every 10th quantile (a total of 9, 10th to 90th).

## RESULTS

In total, 96 participants were recruited, of whom 91 met the selection criteria; 89 participants completed the study: 29 in Experiment 1, 31 in Experiment 2 and 29 in Experiment 3 (Fig. 2). Participants' baseline characteristics were similar between the three experiments. No significant differences in baseline measurements between the three experimental were found. Sociodemographic and baseline measurements are summarised in Table 2.

### Magnitude of the CPM effect

For the linear mixed model analyses, the addition of random slopes did not improve the models, so only random intercepts were used. Table 3 shows the (adjusted) regression coefficients for the design and order effect. Table 4 shows the regression coefficients for the experiment effect and Table 5 presents all estimated marginal means of the CPM and the differences between the designs.

#### *Absolute CPM effect*

The absolute data showed no significant main effect for design or experiments for either the mechanical or thermal test stimulus on the CPM effect (Tables 3 and 4). No interaction (design x experiment) was found either, meaning that the experimental condition did not influence a possible difference between the sequential and parallel design (mechanical (PPT); cervical spine, $p = .626$, forearm, $p = .395$, lower leg, $p = .402$, and for thermal; cervical spine, $p = .540$, forearm, $p = .901$, lower leg, $p = .647$). Figure 3 shows the estimated marginal means of the absolute CPM effects.

#### *Relative CPM effect–Mechanical test stimulus*

Contrary to the absolute data, the relative data showed a significant main effect at the neck for design of $\beta = -6.1\%$ (95% CI [−10.1 to −2.1]; $p = .003$), indicating that the sequential design had a lower CPM effect compared to the parallel design. There was no significant main effect of experiment, indicating that the difference between the experiments in conditioning stimulus intensity (VAS 40 *vs* 60/100), $\beta = 3.5\%$ (95% CI [−2.9 to 9.8]; $p = .282$), or use of dual (mechanical and thermal) *versus* single (mechanical) test stimuli, $\beta = 0.4\%$ (95% CI [−6.0 to 6.7], $p = .908$), did not affect the magnitude of the CPM effect. There was no significant interaction effect between design and experiments, $p = .547$, indicating that the difference between designs was observed in all three experiments

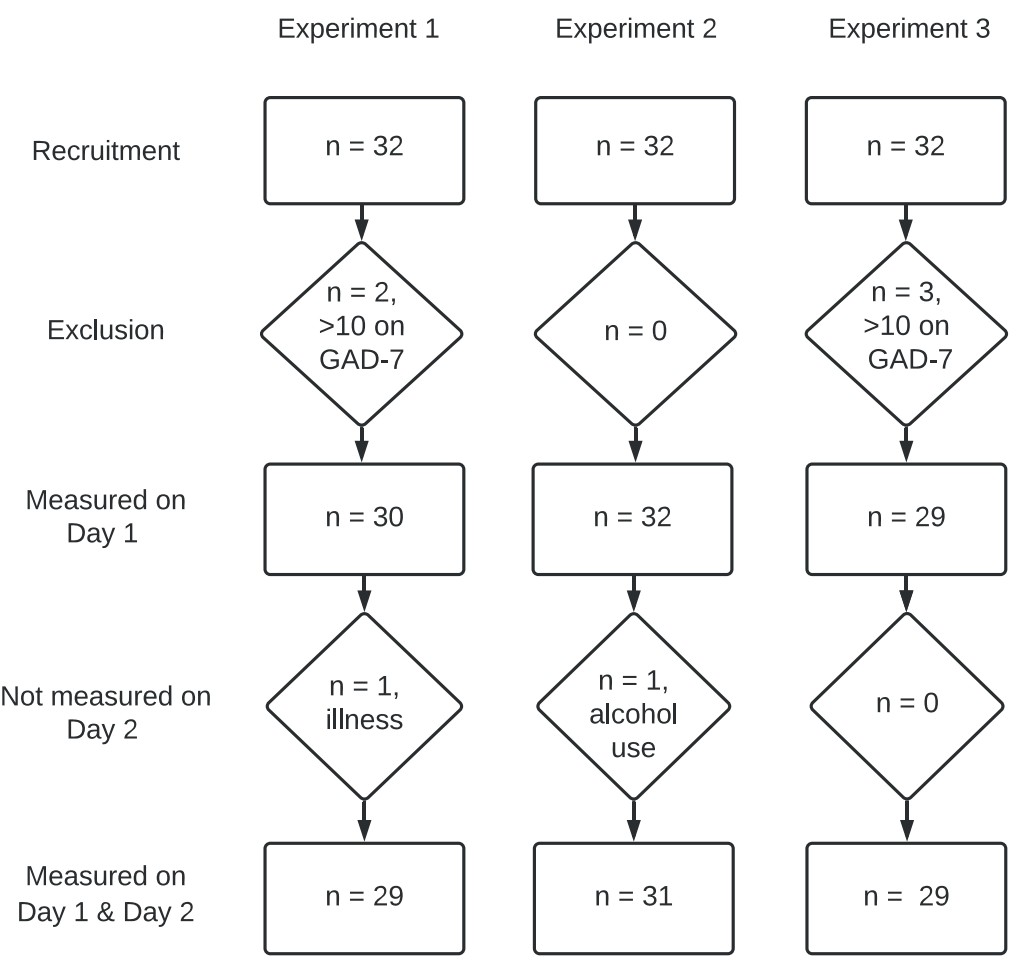

**Figure 2  Flowchart of the study including all three experiments.** Flowchart showing the number of participants within each experiment and reasons for exclusion or lost on the second day. GAD-7, Generalised Anxiety Disorder seven items.

comparably. At the forearm, there was no significant main effect for design, $\beta = -4.5\%$ (95% CI [$-9.0$ to 0.1], $p = .055$), or experiment. Neither the intensity, $\beta = -3.9\%$ (95% CI [$-10.2$ to 2.3], $p = .216$), nor the dual stimulus, $\beta = 0.1\%$ (95% CI [$-6.2$ to 6.3], $p = .984$), had a significant effect. No interaction effect was found between design and experiments, $p = .944$, either. At the lower leg, there was a significant main effect for design, $\beta = -5.9\%$ (95% CI [$-11.7$ to $-0.1$], $p = .046$), indicating a lower CPM effect for the sequential design. No significant main effect of experiment was present, for the comparison of using a lower intensity, $\beta = 5.3\%$ (95% CI [$-2.6$ to 13.3], $p = .185$), or use of a single stimulus, $\beta = 5.2\%$ (95% CI [$-2,8$ to 13.1], 2.8, $p = .197$). There was no interaction effect (design x experiment), $p = .351$.

### Relative CPM effect –Thermal test stimulus
For the heat test stimulus, there was no significant main effect for design at any test location; neck: $\beta = -5.1\%$ (95% CI [$-13.4$ to 3.1], $p = .214$) ; forearm: $\beta = -1.7\%$ (95% CI [$-9.8$

**Table 2  Characteristics of the participants and baseline outcomes.**

| | Experiment 1 Low conditioning stimulus (VAS 40)/ dual test stimuli | Experiment 2 High conditioning stimulus (VAS 60)/ dual test stimuli | Experiment 3 High conditioning stimulus (VAS 60)/ single test stimulus | Total | P-Value[c] |
|---|---|---|---|---|---|
| **Demographics** | n = 29 | n = 31 | n = 29 | N = 89 | |
| Sex, male (%) | 14 (48) | 17 (55) | 14 (48) | 45 (51) | p = .896 |
| Age, years | 24 (21–28)[a] | 24 (22–24)[a] | 22 (20–26)[a] | 24 (21–28)[a] | p = .204 |
| Height, cm | 177 (9) | 181 (10) | 176 (10) | 178 (10) | p = .377 |
| Weight, kg | 75 (12) | 74 (12) | 71 (68–79)[a] | 73 (65–81)[a] | p = .837 |
| BMI, kg/m² | 24.0 (2.9) | 22.5 (2.3) | 23.5 (22.0–25.3)[a] | 23.1 (21.2–25.2)[a] | p = .076 |
| Sports, hours/week | 5 (2.5–6.5)[a] | 4 (3–5)[a] | 4.5 (2.5–6.0)[a] | 4 (3–6)[a] | p = .328 |
| Anxiety, GAD-7 | 1 (0–2.5)[a] | 2 (0–3)[a] | 1 (0–3)[a] | 1 (0–3)[a] | p = .377 |
| **Baseline Pressure Pain Threshold, kPa** | | | | | |
| Neck | | | | | |
| Sequential | 418 (143) | 438 (159) | 456 (142)[b] | 438 (148) | p = .632 |
| Parallel | 410 (127) | 476 (183) | | 448 (154) | p = .242 |
| Forearm | | | | | |
| Sequential | 407 (158) | 398 (138) | 456 (140)[b] | 420 (146) | p = .262 |
| Parallel | 395 (118) | 425 (141) | | 425 (134) | p = .232 |
| Lower leg | | | | | |
| Sequential | 646 (273) | 680 (283) | 727 (277)[b] | 684 (277) | p = .540 |
| Parallel | 577 (496–690)[a] | 705 (276) | | 688 (280) | p = .411 |
| Cold Pressor Test Temperature CS (°C) | 11.9 (11.7–12.0)[a] | 9.0 (8.8–9.3)[a] | 9.9 (9.4–10.7) | | |

**Notes.**
Data are presented as mean (SD) unless otherwise specified.
[a]Median (IQR).
[b]baseline measurements in Experiment 3 are the same for both designs.
[c]No significant differences for sociodemographic data between the three experiments or between sequential versus parallel design BMI, body mass index; GAD-7, generalised anxiety disorder seven items; PPT, Pressure Pain Threshold; kPa, kilo Pascal; CS, Conditioning Stimulus

to 6.3], $p = .670$); lower leg, $\beta = -1.9\%$ (95% CI [$-8.6$ to $4.9$], $p = .586$). Main effects of the experiments were not significant, indicating that a lower intensity of the conditioning stimulus did not yield a significantly different CPM effect, at the neck, $\beta = 4,6\%$ (95% CI [$-3,9$ to $13.2$], $p = .284$), forearm, $\beta = 3.5\%$ (95% CI [$-4.6$ to $11.6$], $p = .392$), or lower leg, $\beta = -0.1\%$ (95% CI [$-10.1$ to $10.0$], $p = .987$). There were no significant interaction effects (design x experiment) observed at the neck, $p = .546$, forearm, $p = .551$, or lower leg, $p = .552$. Figure 4 shows the results of the relative CPM effects.

### The effect of order
Main effects of design were adjusted for order effects, since the sequence in which the designs were tested was randomised. The order effect showed that the design which was performed first, irrespective of which design it was, resulted in a greater CPM effect (23 to 36 kPa for mechanical stimuli, 5 mm VAS (0–100 mm) for thermal testing). This finding was more apparent in the absolute data compared to the relative data (see Table 3).

**Table 3  Effect of Design on magnitude of CPM effect, adjusted for order effect.**

| | | Absolute data | | | | | Relative data (%) | | | | |
|---|---|---|---|---|---|---|---|---|---|---|---|
| | | Design | | | Order | | Design | | | Order | |
| | Location | β, 95% CI | P-Value | f² | β, 95% CI | P-value | β, 95% CI | P-Value | f² | β, 95% CI | P-Value |
| Mechanical | Cervical spine | −3 [−21, 15] | .723 | 0.00 | 24 [6, 43] | **.009** | −6.1 [−10.1, −2.1] | **.003** | 0.10 | 0.1 [−3.9, 4.1] | .967 |
| Pressure Pain Threshold (PPT)[a] | Forearm | 5 [−16, 26] | .633 | 0.00 | 23 [2, 44] | **.034** | −4.5 [−9.0, 0.1] | .055 | 0.04 | −0.7 [−5.3, 3.8] | .745 |
| | Lower leg | −5 [−40, 30] | .780 | 0.00 | 46 [11, 81] | **.010** | −5.9 [−11.7, −0.1] | **.046** | 0.05 | −0.3 [−5.4, 6.1] | .907 |
| Thermal | Cervical spine | −1 [−5, 2] | .501 | 0.01 | 3 [−1, 6] | .120 | −5.1 [−13.4, 3.1] | .214 | 0.03 | 6.3 [−1.9, 14.5] | .129 |
| Heat Test Stimulus (HTS)[b] | Forearm | −1 [−4, 1] | .324 | 0.01 | 3 [0, 6] | .060 | −1.7 [−9.8, 6.3] | .670 | 0.00 | 5.8 [−2.2, 13.9] | .156 |
| | Lower leg | 1 [−2, 3] | .579 | 0.01 | 5 [2, 7] | **<.001** | −1.9 [−8.6, 4.9] | .586 | 0.00 | 9.2 [2.4, 15.9] | **.009** |

**Notes.**

Significant effects are printed in bold. Test location at the cervical spine is paraspinal muscles over C5/C6, at the forearm is extensor carpi radialis muscle and at the lower leg is the tibialis anterior muscle. A negative coefficient for design indicates the sequential design is lower compared to the parallel design. Order effect is based upon if the design was either first or second. A negative coefficient for order indicates that the first measurement is lower than the second measurement.

[a] Absolute data in kPa.
[b] Absolute data in VAS (0–100).

**Table 4  Main effect of experiments on magnitude of CPM effect.**

| | | Absolute data | | | | Relative data (%) | | | |
|---|---|---|---|---|---|---|---|---|---|
| | | Experiment 1 | | Experiment 3 | | Experiment 1 | | Experiment 3 | |
| | Location | β, 95% CI | P-value | β, 95% CI | P-Value | β, 95% CI | P-Value | β, 95% CI | P-Value |
| Mechanical | Cervical spine | 17 [−15, 48] | .296 | −21 [−53, 11] | .189 | 3.5 [−2.9, 9.8] | .282 | 0.4 [−6.0, 6.7] | .908 |
| Pressure Pain Threshold (PPT)[a] | Forearm | −19 [−52, 14] | .260 | 19 [−52; 14] | .252 | −3.9 [−10.2, 2.3] | .216 | 0.1 [−6.2, 6.3] | .984 |
| | Lower leg | 47 [−14, 108] | .132 | −74 [−135, −13] | **.018** | 5.3 [−2.6, 13.3] | .185 | 5.2 [−2.8, 13.1] | .197 |
| Thermal | Cervical spine | 1 [−2, 5] | .476 | – | – | 4.6 [−3.9, 13.2] | .284 | – | – |
| Heat Test Stimulus (HTS)[b] | Forearm | 0 [−3, 3] | .773 | – | – | 3.5 [−4.6, 11.6] | .392 | – | – |
| | Lower leg | −1 [−4, 1] | .321 | – | – | −0.1 [−10.1, 10.0] | .987 | – | – |

**Notes.**

Significant effects are printed in bold. Test location at the cervical spine is paraspinal muscles over C5/C6, at the forearm is extensor carpi radialis muscle and at the lower leg is the tibialis anterior muscle. Experiment 1; both test stimuli, conditioning stimulus VAS 40/100, Experiment 2 (control); both test stimuli, conditioning stimulus VAS 60/100, Experiment 3; only mechanical test stimulus, conditioning stimulus VAS 60/100. Coefficients shown in comparison with Experiment 2 as control, resulting in the coefficients of Experiment 1 reflecting the difference in the intensity of the conditioning stimulus and the coefficients of Experiment 3 reflecting the difference of single or dual test stimuli. Thermal testing was absent in Experiment 3.

[a] Absolute data in kPa.
[b] Absolute data in VAS (0 −100).

### *Differences in effect of design between higher and lower levels of CPM*

Quantile regression analyses (10th till 90th quantile) revealed no significant different regression coefficients compared to the main analyses, indicating that in pro-nociceptive up to anti-nociceptive CPM effects, no different main effects for design were found. Figure 5 shows the estimated regression coefficients (adjusted for order) and 95% confidence intervals for the 10th till the 90th quantiles including the estimated regression coefficient

**Table 5 Estimated marginal means of the of CPM effect (adjusted for order effect) in a sequential versus parallel design.**

| Pressure Pain Threshold (PPT) | | Absolute CPM effect (kPa, 95% CI) | | | Relative CPM effect (%, 95% CI) | | |
|---|---|---|---|---|---|---|---|
| | | Sequential | Parallel | Difference | Sequential | Parallel | Difference |
| Cervical spine | Overall | 1 [−15, 17] | 4 [−11, 20] | −3 [−21, 15] | −2.4 [−5.7, 0.9] | 3.7 [0.4, 7.0] | **−6.1 [−10.1, −2.1]**[a] |
| | Experiment 1 | 2 [−19, 24] | 12 [−9, 33] | −10 [−37, 17] | 0.0 [−5.1, 5.1] | 5.8 [0.6, 10.9] | −5.8 [−11.9, 0.4] |
| | Experiment 2 | −10 [−34, 15] | −9 [−34, 15] | −0 [−35, 35] | −3.1 [−8.8, 2.7] | 1.9 [−3.9, 7.7] | −5.0 [−13.1, 3.2] |
| | Experiment 3 | 11 [−24, 47] | 12 [−24, 47] | −0 [−31, 30] | −4.1 [−10.3, 2.2] | 3.6 [−2.6, 9.9] | **−7.7 [−12.8, −2.6]**[a] |
| Forearm | Overall | 12 [−6, 29] | 7 [−11, 24] | 5 [−16, 26] | −1.2 [−4.7, 2.3] | 3.3 [−0.2, 6.7] | −4.5 [−9.1, 0.2] |
| | Experiment 1 | −5 [−30, 19] | −14 [−39, 10] | 9 [−25, 44] | −3.8 [−9.2, 1.6] | 0.5 [−4.9, 5.9] | −4.3 [−11.9, 3.3] |
| | Experiment 2 | 16 [−9, 40] | 2 [−22, 27] | 13 [−22, 48] | 0.1 [−6.2, 6.0] | 4.7 [−1.4, 10.8] | −4.8 [−13.4, 3.8] |
| | Experiment 3 | 24 [−15, 64] | 32 [−7, 72] | −8 [−39, 23] | 0.2 [−6.3, 6.6] | 4.5 [−1.9, 11.0] | −4.4 [−9.4, 0.6] |
| Lower leg | Overall | 42 [11, 73] | 47 [16, 78] | −5 [−40, 30] | 4.2 [−0.2, 8.6] | 10.1 [5.7, 14.5] | **−5.9 [−11.8, −0.0]**[a] |
| | Experiment 1 | 53 [6, 99] | 52 [5, 98] | 1 [−62, 64] | 5.2 [−2.6, 13.0] | 13.0 [5.1, 20.8] | −7.7 [−18.5, 3.0] |
| | Experiment 2 | 10 [−43, 62] | 1 [−52, 53] | 9 [−64, 82] | 2.7 [−5.8, 11.3] | 4.8 [−3.8, 13.3] | −2.0 [−14.1, 10.1] |
| | Experiment 3 | 66 [6, 127] | 92 [32, 153] | −26 [−73, 20] | 4.8 [−1.3, 10.8] | 13.1 [7.0, 19.2] | **−8.4 [−13.0, −3.8]**[a] |

| Heat Test Stimulus (HTS) | | Absolute CPM effect (VAS 0–100, 95% CI) | | | Relative CPM effect (%, 95% CI) | | |
|---|---|---|---|---|---|---|---|
| | | Sequential | Parallel | Difference | Sequential | Parallel | Difference |
| Cervical spine | Overall | 3 [0, 6] | 4 [2, 7] | −1 [−5, 2] | 5.2 [−0.7, 11.1] | 10.4 [4.5, 16.3] | −5.1 [−13.4, 3.1] |
| | Experiment 1 | 4 [1, 8] | 4 [1, 8] | −0 [−5, 5] | 9.0 [1.0, 17.0] | 11.4 [3.4, 19.4] | −2.4 [−13.7, 8.9] |
| | Experiment 2 | 2 [−2, 5] | 4 [1, 8] | −2 [−7, 2] | 1.5 [−7.0, 10.1] | 9.6 [1.0, 18.2] | −8.1 [−18.7, 2.5] |
| Forearm | Overall | 2 [−0, 4] | 3 [1, 5] | −1 [−4, 1] | 2.9 [−2.8, 8.6] | 4.7 [−1.0, 10.3] | −1.7 [−9.8, 6.3] |
| | Experiment 1 | 2 [−1, 5] | 3 [1, 6] | −1 [−5, 3] | 3.6 [−4.0, 11.2] | 7.6 [0.0, 15.2] | −4.1 [−14.8, 6.7] |
| | Experiment 2 | 1 [−1, 4] | 3 [1, 6] | −2 [−5, 2] | 2.2 [−6.2, 10.6] | 2.0 [−6.4, 10.4] | −0.1 [−10.6, 10.9] |
| Lower Leg | Overall | 2 [−2, 4] | 3 [1, 5] | −1 [−4, 1] | 2.8 [−3.2, 8.8] | 5.4 [−0.1, 11.4] | −2.6 [−9.3, 4.1] |
| | Experiment 1 | 1 [−1, 3] | 2 [−0, 4] | −1 [−2, 1] | 1.7 [−7.3, 10.6] | 6.5 [−2.5, 15.4] | −4.8 [−14.0, 4.4] |
| | Experiment 2 | 3 [0, 6] | 2 [−1, 5] | 1 [−3, 5] | 3.9 [−4.3, 12.1] | 4.4 [−3.8, 12.6] | 0.1 [−10.4, 9.4] |

**Notes.**
Estimated marginal means are adjusted for order effect. Significant differences are printed in bold. Relative data of 0.0% indicates that the baseline test and conditioned test were the same; no CPM effect is present. Positive values reflect an inhibitory effect for both absolute and relative data. Test location at the cervical spine is paraspinal muscles over C5/C6, at the forearm is extensor carpi radialis muscle and at the lower leg is the tibialis anterior muscle. CPM, Conditioned Pain Modulation; kPa, kilo Pascal; VAS, Visual Analogue Scale.

[a] $p < .05$

and 95% confidence interval based on linear mixed model regression (main analysis) per outcome and location.

## DISCUSSION

This study in healthy, young participants demonstrated that the magnitude of the CPM effect is not significantly different between a parallel and sequential design. Only when calculating the relative CPM effect, minimal, but no relevant higher effects were found for the parallel design compared to the sequential design. All tested locations showed CPM effects of comparable magnitudes. Additionally, experimental conditions, such as pain

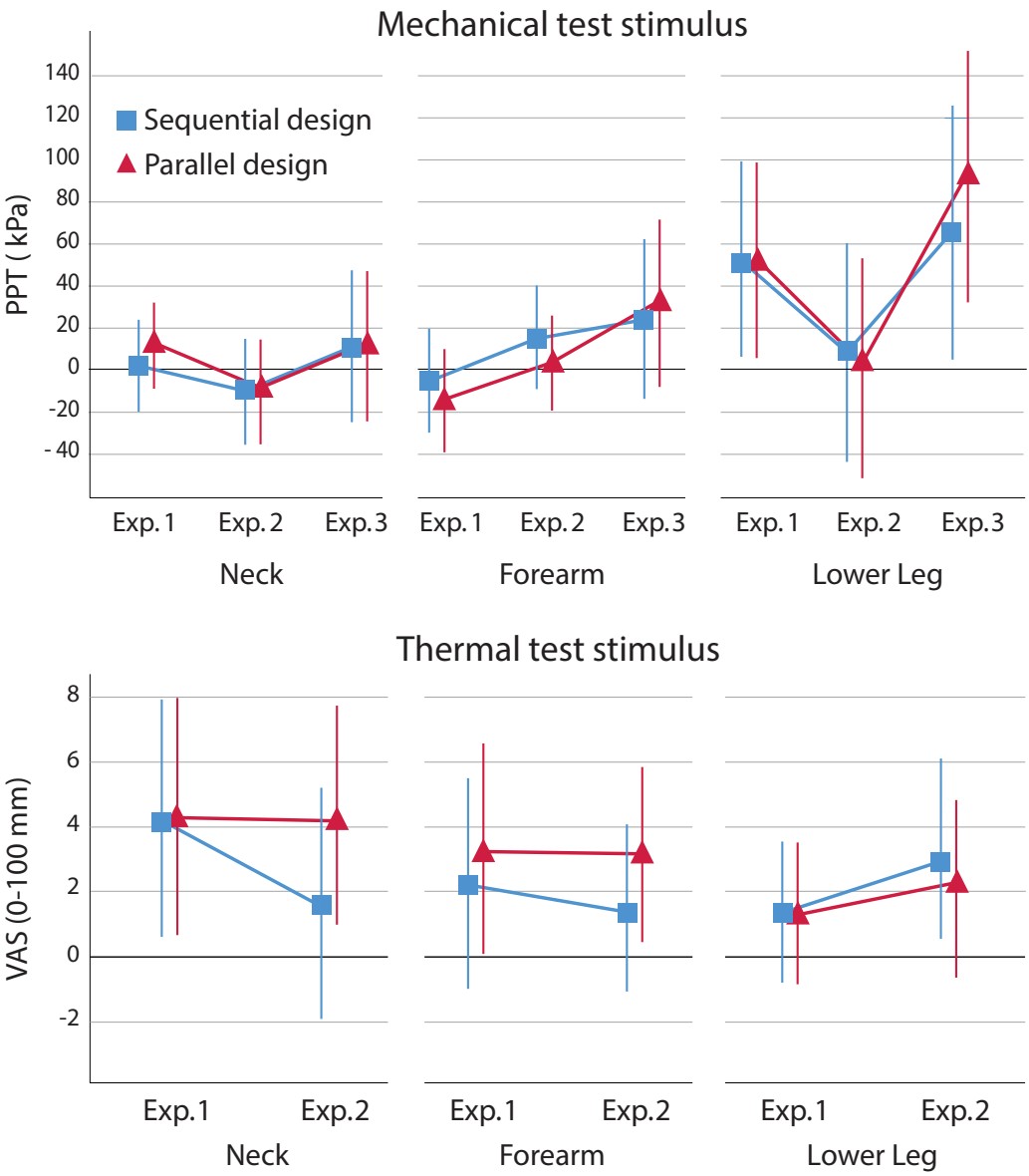

**Figure 3 Absolute CPM effects.** Absolute CPM effect based on estimated marginal means with a mechanical and thermal test stimulus for the sequential (blue) and parallel (red) design per experiment, including 95% confidence intervals, presented per location. Positive values reflect an inhibitory effect and negative values a facilitatory effect.

# Relative CPM effect

## Mechanical test stimulus

## Thermal test stimulus

**Figure 4  Relative CPM effects.** Relative CPM effect based on estimated marginal means with a mechanical and thermal test stimulus for the sequential (blue) and parallel (red) design per experiment, including 95% confidence intervals, presented per location. Positive values reflect an inhibitory effect and negative values a facilitatory effect.

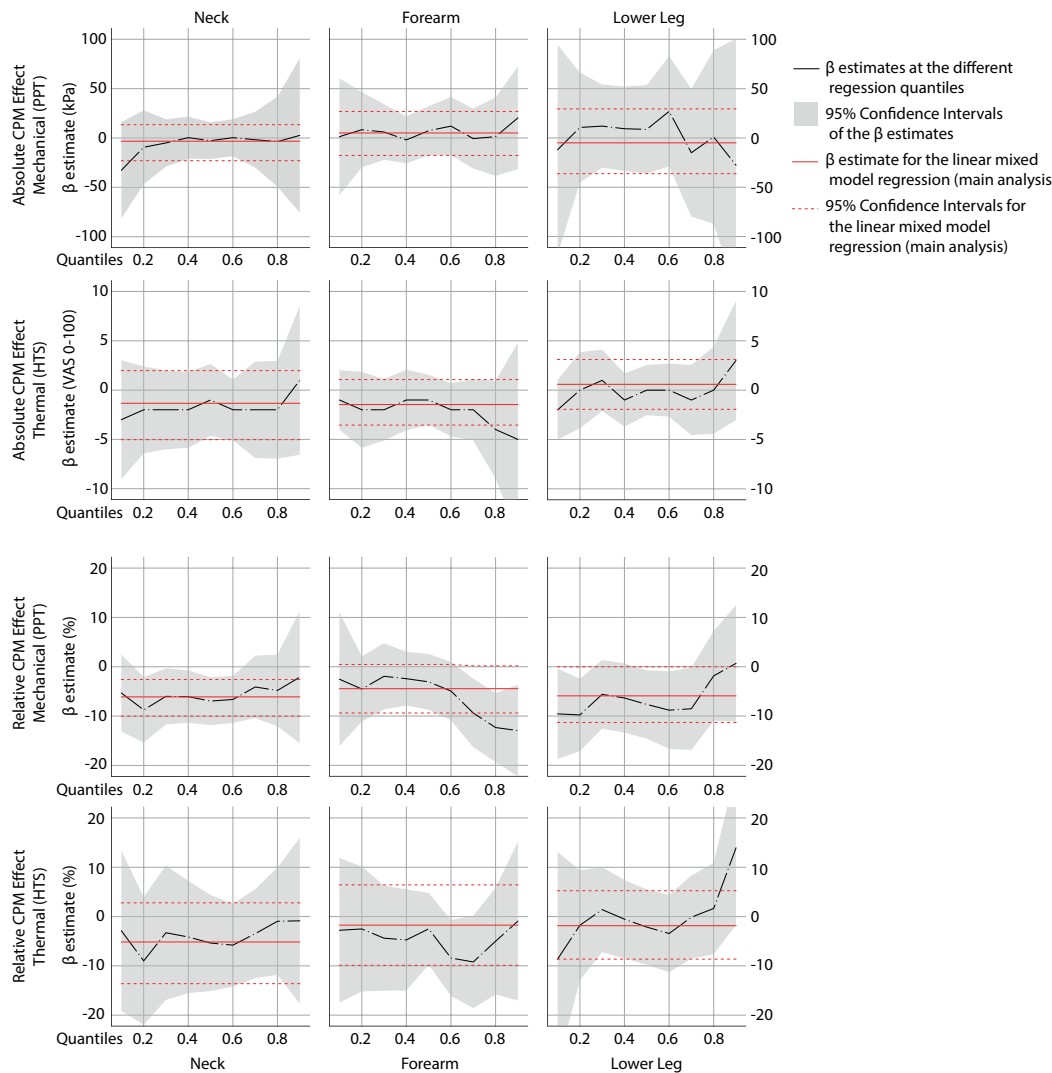

**Figure 5   Quantile regression plots.** Quantile regression plots with estimated regression coefficient for the main effect of design, per 10th quantile, including 95% confidence intervals and regression estimate based on linear mixed model regression (main analysis), presented per outcome and test location. A negative coefficient indicates that the sequential design has a lower effect compared to the parallel design.

intensity (VAS 40 and 60/100) of the conditioning stimulus, or using a single (mechanical) *versus* dual (mechanical and thermal) test stimuli, showed no significant influence on either the differences in the designs or on the magnitude of the CPM effect itself.

The slightly higher relative CPM effects in the parallel design may partly be attributed to the substantially longer duration of the conditioning stimulus compared to the sequential design. However, previous research showed conflicting results on the relation between duration of the conditioning stimulus and magnitude of the CPM effect (*Graven-Nielsen et al., 2017*; *Razavi et al., 2014*; *Smith & Pedler, 2018*). One study found a higher CPM effect with a longer duration (6–12 min) of contact heat conditioning stimulus compared

to a shorter duration (0–6 min), but only with a high intensity of the conditioning stimulus (*Razavi et al., 2014*). Contrary to other studies, no significant differences in CPM effect were found when using 10 s duration of cuff occlusion *versus* 60 s, and 20 s *versus* 3 min (*Graven-Nielsen et al., 2017*; *Smith & Pedler, 2018*).

Published recommendations advise a sequential design due to potentially less distraction or attention bias (*Yarnitsky et al., 2015*). Attention, away from a pain-inducing stimulus, has a general analgesic effect and might increase the measured CPM effect (*Hoegh, Seminowicz & Graven-Nielsen, 2019*; *Moont et al., 2012*; *Moont et al., 2010*). Additionally, focussing attention on the conditioning stimulus gives a more potent inhibition compared to focussing attention on the test stimulus (*Ladouceur et al., 2012*; *Mertens et al., 2020*). It is therefore possible that attention bias has caused a slight overestimation of the CPM-effect in our parallel design, when using a mechanical test stimulus. Interestingly, we found no differences when using the thermal test stimulus, where we asked the participants to rate the pain intensity every 10 s, possibly due to increased focus on the test stimulus. Consequently, this could have decreased the influence of attention bias on the conditioned stimulus leading to a more comparable construct as sequential testing. Further, since the time in the cold water bath is considerably shorter using the sequential design (1 min) compared to the parallel design (5–7 min), this may lead to potentially less attention bias and less burden to participants.

The effect of design was in most analyses smaller than the effect of order. The magnitude of the CPM effect was slightly higher during the first measurement compared to the second measurement, which could be explained by a habituation or salience effect due to the repetitive nature of the study (*Hall & Rodríguez, 2017*). These observations are in line with a recent study which tested the CPM effect two days after the first measurement in people with experimentally-induced low back pain and found decreased CPM values on the second test occasion (*McPhee & Graven-Nielsen, 2018*). Additionally, the first measurements influence the participant's expectation, which can influence the measurements on the second day (*Goffaux et al., 2007*).

Differences in relative and absolute CPM effects can be explained by their different calculations. Relative CPM effects use the unconditioned test (*i.e.,* baseline PPT or heat threshold) as basis for the calculation. The outcome reflects the individual's pain inhibition, since it is corrected for their baseline test result. For example, in people with low PPT or heat baseline thresholds, small changes will lead to small absolute CPM effects, but can result in high relative CPM effects as this change is compared to the individual's baseline. We observed this effect on the group outcome (*i.e.,* in the neck), where differences in designs were only observed in the relative CPM effects and not in absolute CPM effects.

Our results showed no significant difference in CPM effect when different intensities of the conditioning stimulus were used. We used VAS 40/100 (Experiment 1) and VAS 60/100 (Experiment 2 and 3) pain levels, in line with Nir et al. (*Nir et al., 2011*), who also found no differences in magnitude of the CPM effect between these two intensities. Although no direct comparison can be made, higher CPM effects were observed when using a much colder water bath (∼3 °C (*Skovbjerg et al., 2017*) *versus* ∼12 °C in Experiment 1; and ∼9 °C in Experiment 2 and 3). This may be explained by the experienced pain intensity using

a ~3 °C water bath, which is probably outside the moderate-intensity range used in our experiments. A high-intensity may have a different effect then moderate-intensity and the differences between our intensities were not large enough to yield results.

Although previous research found high anti-nociceptive CPM effects in healthy participants (*Lewis, Rice & McNair, 2012*; *Skovbjerg et al., 2017*), an increasing number of studies find, in line with our study, low mean CPM effects and large intra-individual variability (*Klyne et al., 2018*; *Larsen, Madeleine & Arendt-Nielsen, 2019*; *Owens et al., 2016*; *Vaegter et al., 2018*). One study with young, healthy adults found a mean CPM magnitude of ~0 kPa, similar to our results (*Klyne et al., 2018*). Their sample consisted of two groups, young patients with low back pain and young, healthy controls. These authors identified factors which were significantly associated with the absence of the CPM effect: frequency and amount of alcohol consumption, and sleep disturbance and sleep latency. These factors, although not measured, can be anticipated to be also present in our participants as the setting where the participants were recruited was comparable. These factors seem to influence the CPM effect more than age, as previous research showed that the CPM effect should be higher in younger people (*Hackett, Naugle & Naugle, 2020*; *Skovbjerg et al., 2017*). Future research is required to explore factors explaining the variability of the magnitude of the CPM effect.

A repeated-measurement design with relatively long within-session rest periods could have resulted in a loss of attention of our participants, especially for the last measurements of the sessions. Therefore, we randomised the order of measurements (both test stimuli and both designs) and statistically corrected for the order effect, but this could have resulted in lower mean group CPM effects. The low mean CPM effects could potentially masque differences in the designs.

Additional quantile regression analyses, however, showed no significantly different results in estimated regression coefficient distribution, meaning that even in participants with higher (anti-nociceptive) or lower (pro-nociceptive) CPM effects, no significant differences between the sequential and parallel design were present.

Furthermore, we tried to reduce the influences of confounders by using a highly standardised protocol to control for variable conditions, including standardised communication with the participants and using strict selection criteria and balanced recruitment for age and sex. The strictness of the protocols used, may have reduced the generalisability of the measurements into clinical practice but was beneficial for the study's aim.

## CONCLUSIONS

This study compared a sequential and parallel design and found no relevant differences in CPM effects between the two designs. No effect of conditioning stimulus intensity or the use of dual test stimuli on a potential difference was found. Since the small differences in CPM effects were only found in the relative data, it may be of importance to analyse and report both absolute and relative data in future research, as reported in the recommendations for measuring CPM effects.

## ACKNOWLEDGEMENTS

We would like to thank all participants and research assistants, in particular Caspar Mylius, for their assistance with the project.

### Funding

This study was conducted with a research grant for teachers of the Dutch Organisation of Scientific Research (NWO) under project number 023.011.069. The funders had no role in study design, data collection and analysis, decision to publish, or preparation of the manuscript.

### Grant Disclosures

The following grant information was disclosed by the authors:
The Dutch Organisation of Scientific Research (NWO) under project number 023.011.069.

### Competing Interests

The authors declare there are no competing interests.

### Author Contributions

- Roland R. Reezigt conceived and designed the experiments, performed the experiments, analyzed the data, prepared figures and/or tables, authored or reviewed drafts of the paper, and approved the final draft.
- Sjoerd C. Kielstra conceived and designed the experiments, performed the experiments, authored or reviewed drafts of the paper, and approved the final draft.
- Michel W. Coppieters and Gwendolyne G.M. Scholten-Peeters conceived and designed the experiments, prepared figures and/or tables, authored or reviewed drafts of the paper, and approved the final draft.

### Human Ethics

The following information was supplied relating to ethical approvals (i.e., approving body and any reference numbers):

The study was approved by the scientific and ethical review board of the Vrije Universiteit Amsterdam, The Netherlands (VCWE-2017-022R1).

### Data Availability

The raw measurements are available in the Supplementary File after de-identification of the participants.

### Supplemental Information

Supplemental information for this article can be found online at http://dx.doi.org/10.7717/peerj.12330#supplemental-information.

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
