# Peer review of "No relevant differences in conditioned pain modulation effects between parallel and sequential test design. A cross-sectional observational study"

_PeerJ, doi:10.7717/peerj.12330_

## Round 0.1 · original submission · Major Revisions

Reviewers found merit in your manuscript, however, both offered suggestions for improvement. In particular, the reviewers found some parts of the design/manuscript difficult to follow and have offered suggestions for greater clarity.

Reviewer 1 ·

Basic reporting

Basic Reporting
Abstract:
1. Line 34-35, “either during (parallel design) or after (sequential design) the test stimulus” – do you mean either during or after the conditioning stimulus? This might be confusing for individuals who are not familiar with CPM.
2. The methods section of the abstract is currently lacking statistical analyses. This may be due to a limited word count, but if it allows, I recommend the authors add statistical analyses that used in the present study.

Introduction:
3. Line 65, please provide examples of mechanical and thermal test stimuli.
4. Line 67, please be more specific with the test and conditioning stimuli. For example, CPM consist of two test stimuli, and conditioning stimuli can be applied consecutively or simultaneously with the second test stimulus but not with the first test stimuli.
5. Line 71-72, the last sentence of this paragraph seems unnecessary.
6. Paragraph 3-5, I think it would be of benefit to make them into one single paragraph rather than having multiple paragraphs.
7. Line 85, please explain further what are the “conflicting results”.
8. Line 95-96, this paragraph seems unnecessary. I suggest deleting it or combining with the previous paragraph.
9. Line 100-101, study aim 2) is somewhat confusing. Please address it more clearly.

Results:
10. In Figure 2, one of “Experiment 2” has to be “Experiment 3”.
11. In Table 2, what do you mean by “baseline measurements in Experiment 3 are the same for both designs”? It was mentioned that both a sequential and parallel design were performed in each experiment. Please explain why there is only one baseline PPT value for Experiment 3.
12. Please double-check and match all the values (e.g. β, 95% CI values) in the manuscript with those values in the tables to avoid any confusion.
13. Please address the results of the order effect (Table 3) in the results part. Although it is briefly explained in Table 3 and discussion, it might be helpful for reader to further understand what those results represent.

References:
14. Please double-check the reference list (e.g. reference #83-84, #92-95).

Experimental design

Methods:
15. Were participants familiarized with the procedures on the first testing day or on a separate day? Please provide it in the familiarization part of the methods.
16. Line 134, please provide how long did participants rest between parallel and sequential design for Experimental 3. Since Experimental 3 was performed in a single day whereas other two experiments were performed on two consecutive days, is it problematic to compare these three experiments for the first study aim?
17. Line 152, were participants also refrained from exercise? If not, is there any possibility that exercise could influence the CPM effect (e.g. exercise-induced hypoalgesia).
18. Was PPT measured first and then followed by thermal stimuli? It is mentioned that there was 10 minutes between test stimuli, however, the order of test stimuli is not stated.
19. Line 286-288, please further explain what “regression coefficients adjusted for the effect of order of measurements (first or second measurement)” reflects or means (e.g. why does it have to be adjusted and what does it tell us?).

Validity of the findings

Discussion & Conclusion:
20. Line 417-425, this paragraph includes useful information, however, this paragraph appears to be unrelated to your main research questions.
21. Line 426-434, please provide rationale for why both absolute and relative values were reported in the present study (e.g. is it recommended to report both absolute and relative values? If so, why is it recommended?).
22. Higher relative CPM effects (PPT) in the parallel design were observed at cervical spine and lower leg, but not at forearm. It might be useful to address and explain these findings.
23. As CPM effect was examined only at 3 sites (i.e., CS, forearm, lower leg), this should not be generalized. Once again, the authors did not observe higher relative CPM effects in the parallel design at forearm. This may indicate that the influence of design on CPM effects may differ between limbs or sites. I think it would be of benefit to address this point in discussion.

Additional comments

Research questions of this study are interesting. However, the manuscript was not easy to follow. As an overall comment, I would strongly encourage the authors to improve the readability of the paper (also for those readers who may not be familiar with the CPM literature).

·

Basic reporting

The manuscript is well-written with clear English language throughout. A significant body of literature has been acknowledged and discussed throughout. It is evident that the authors have a deep understanding of the existing literature and gaps in the research area. Referencing, structure and tables are appropriate.

Experimental design

The research questions are relevant to the area of research and are appropriate to address the knowledge gap identified. These questions are designed to deliver important, relevant data within the field. Due to the study design (i.e. multiple aims across multiple experiments), I have provided the authors with a suggestion for improving the clarity of these aims. The methodology is very good, with exceptional detail provided within the manuscript. It is clear that this was a well-controlled experiment.

Validity of the findings

Raw data is provided, and the analysis methods are sound. The conclusions are supported by the data and are discussed relative to the research questions and existing literature.

Additional comments

Overall this is an interesting study that targets an important area of pain research. The authors should be commended for the design and control of the study, the thorough analysis conducted and the overall quality of the written manuscript. It is clear the authors have a deep understanding of the research area. I have two suggestions that I hope will add additional strength to the manuscript, and have detailed some specific revisions to be made.

General comments

If possible, please include effects sizes for your comparisons.

The description of how each of the study aims were addressed across the different experiments is sufficiently detailed but may appear confusing to some readers. Presentation of this information in figure format would be welcome; please consider this.

Lines 70-71 – Please consider expanding upon this point, to provide detail on/summarise the discrepancies within the literature so it is clear for the reader. This would add further support for the study rationale and some of the discussion points following the results.


Specific comments

Line 38 – ...as “the” conditioning stimulus?

Line 61 – please add a space between the last word and references

Line 68 – add a space between the period and beginning of the next sentence

Line 85 – remove hyphen

Lines 92 and 94 - please add a space between the last word and reference

Line 95-96 – This paragraph is one sentence long. Please merge with the previous paragraph.

Line 402 = “advise”

---

## Round 0.2 · accepted · Accept

Both reviewers found your revision acceptable for publication.

Reviewer 1 ·

Basic reporting

no comment

Experimental design

no comment

Validity of the findings

no comment

·

Basic reporting

No additional comment

Experimental design

No additional comment

Validity of the findings

No additional comment

Additional comments

No additional comment